# Deep-Learning-Based High-Intensity Focused Ultrasound Lesion Segmentation in Multi-Wavelength Photoacoustic Imaging

**DOI:** 10.3390/bioengineering10091060

**Published:** 2023-09-08

**Authors:** Xun Wu, Jean L. Sanders, M. Murat Dundar, Ömer Oralkan

**Affiliations:** 1Department of Electrical and Computer Engineering, North Carolina State University, Raleigh, NC 27606, USA; jlsander@umich.edu; 2Computer and Information Science Department, Indiana University—Purdue University, Indianapolis, IN 46202, USA; mdundar@iupui.edu

**Keywords:** multi-wavelength photoacoustic imaging, high-intensity focused ultrasound therapy, lesion segmentation, deep learning, machine learning, convolutional neural network

## Abstract

Photoacoustic (PA) imaging can be used to monitor high-intensity focused ultrasound (HIFU) therapies because ablation changes the optical absorption spectrum of the tissue, and this change can be detected with PA imaging. Multi-wavelength photoacoustic (MWPA) imaging makes this change easier to detect by repeating PA imaging at multiple optical wavelengths and sampling the optical absorption spectrum more thoroughly. Real-time pixel-wise classification in MWPA imaging can assist clinicians in monitoring HIFU lesion formation and will be a crucial milestone towards full HIFU therapy automation based on artificial intelligence. In this paper, we present a deep-learning-based approach to segment HIFU lesions in MWPA images. Ex vivo bovine tissue is ablated with HIFU and imaged via MWPA imaging. The acquired MWPA images are then used to train and test a convolutional neural network (CNN) for lesion segmentation. Traditional machine learning algorithms are also trained and tested to compare with the CNN, and the results show that the performance of the CNN significantly exceeds traditional machine learning algorithms. Feature selection is conducted to reduce the number of wavelengths to facilitate real-time implementation while retaining good segmentation performance. This study demonstrates the feasibility and high performance of the deep-learning-based lesion segmentation method in MWPA imaging to monitor HIFU lesion formation and the potential to implement this method in real time.

## 1. Introduction

High-intensity focused ultrasound (HIFU) is a minimally invasive technique for in situ tissue destruction [1]. In HIFU ablation, ultrasound (US) waves are focused at a target within the human body to deliver and deposit acoustic energy, causing tissue destruction through thermal and mechanical effects [2]. The thermal effect causes a local temperature rise that leads to irreversible cell death through coagulative necrosis. The mechanical effect, also called cavitation, causes irreversible damage by mechanically disrupting cell membrane permeability and altering the structure of cells [3]. In the past several decades, HIFU has been clinically used to treat bone, breast, kidney, liver, pancreas, and prostate cancer [4,5,6,7,8,9]; uterine fibroids [10]; and cardiac arrhythmias [11,12,13]. To ensure safe and precise ablation, a reliable imaging modality is needed to guide and monitor the HIFU therapy [14]. Two imaging modalities, magnetic resonance imaging (MRI) and US imaging, are currently used for HIFU monitoring [15].

MRI has been used to monitor HIFU for breast fibroadenoma and uterine fibroids [16,17]. It provides superior depiction of anatomic detail and superb real-time thermometry [14,18]. Therefore, it allows precise spatial assessment of the tissue and is the only imaging technique in clinical practice that provides quantitative temperature measurements [19]. However, MRI is limited by its high cost and relatively low frame rate [20,21]. Compared with US-based imaging modalities, MRI equipment is bulkier and less compatible with HIFU therapies.

While MRI mainly monitors the thermal effect caused by HIFU, US imaging focuses more on the mechanical effect, which produces a hyperechoic spot at the focal point of the HIFU transducer [22]. This real-time and low-cost imaging modality can be easily integrated into an HIFU system [23] and has been used to monitor HIFU therapies for prostate cancer [24], breast fibroadenoma [25], and uterine fibroids [26,27]. However, the biological and physical mechanisms of the hyperechoic depiction in US images are not yet fully understood [28], which makes US imaging less reliable for HIFU monitoring. Furthermore, US imaging has relatively low-imaging contrast, sensitivity, and specificity for non-invasive detection [29]. Although US imaging can also provide thermometry by measuring local changes in speed of sound, its sensitivity and accuracy of thermometry are relatively low becuase of the complex physical nature of acoustic tissue properties and mechanisms of cavitation [3].

Since both thermal and mechanical effects fade gradually after the cessation of HIFU exposure, an imaging modality that focuses on monitoring the permanent tissue property change caused by HIFU could greatly supplement or even replace MRI and US imaging. Photoacoustic (PA) imaging is a hybrid imaging modality that combines the contrast of optical absorption with the resolution and penetration depth of US imaging [30,31,32,33]. This imaging modality can be used to monitor HIFU therapies because HIFU induces changes in the optical absorption spectrum of the ablated tissue [34]. However, these changes could be too subtle to be observed because the optical absorption spectrum does not change very significantly at some arbitrarily chosen wavelength [35]. Multi-wavelength photoacoustic (MWPA) imaging generates a multi-channel PA image with each channel corresponding to a specific wavelength. This makes it easier to detect the changes in optical absorption spectrum since changes in tissue absorption can be observed at multiple wavelengths. An initial study on the potential of MWPA imaging in revealing HIFU-induced tissue denaturation was reported in 2013 [36], followed by a study in 2014 [34] that demonstrated the feasibility of MWPA imaging in detecting HIFU-induced thermal lesions. In 2018, Iskander-Rizk et al. proposed a lesion detection algorithm based on dual-wavelength PA imaging [37]. These studies demonstrated the feasibility of detecting HIFU lesions with MWPA imaging. However, none of these studies provided an approach to segment the lesions in MWPA images. A correlation-based approach to segment HIFU lesions was reported in 2016 [38]. In this approach, each pixel was classified based on its Pearson correlation values with the ablated and non-ablated reference spectra, respectively. The ablated reference spectrum is obtained by averaging pixel values inside a region of interest (ROI), which is selected based on the PA image contrast. However, this mean value is not a comprehensive representation of the averaged pixels, especially when their variance is not small. The non-ablated reference spectrum is obtained by normalizing the known extinction spectrum for deoxyhemoglobin (Hb). However, the actual non-ablated reference spectra could be different. This approach does not extract “deeper” features that could be more effective in pixel classification than the Pearson correlation values.

Semantic segmentation is a task of simultaneous object recognition and pixel-wise classification [39]. The past eight years have witnessed the rapid development of convolutional neural network (CNN) as a novel approach for semantic segmentation [40,41,42,43,44,45]. Compared with the per-pixel classification approaches that process each pixel independently, CNN captures the 2D textural features in the MWPA images in addition to the optical absorption spectrum. It therefore has potentially better performance in lesion segmentation. In this paper, we propose a CNN-based approach for HIFU lesion segmentation in MWPA images. HIFU is applied on ex vivo bovine tissue to create HIFU lesions. The bovine tissue is then imaged with MWPA imaging. The acquired MWPA images are processed with a CNN to segment the HIFU lesions. To evaluate the performance of this approach more comprehensively, traditional machine learning algorithms including quadratic discriminant analysis (also called quadratic classifier) [46], fully connected neural network [47], support vector machine [48], and random forest [49] are also trained and tested, and the performance is compared with that of the CNN. Feature selection is performed to reduce the number of wavelengths to facilitate real-time implementation while keeping an acceptable performance. The target application for this approach is catheter-based cardiac ablation with a capacitive micromachined ultrasonic transducer (CMUT) array that can be used for both imaging and ablation [50], where the lesion depth is up to 8 mm, which is comparable to the depth of radiofrequency ablation [51]. The experiment for MWPA data acquisition and the preprocessing of MWPA images are described in detail in Section 2. The CNN-based approach and the traditional machine learning algorithms are also introduced in Section 2, followed by a comparison of their performances and an analysis in Section 3. Section 4 discusses the advantages and disadvantages of the presented approach and the potential to operate it in real time.

## 2. Materials and Methods

### 2.1. Data Acquisition and Image Preprocessing

#### 2.1.1. Experimental Setup

A programmable imaging system (Vantage 64 LE, Verasonics Inc., Kirkland, WA, USA), a personal computer (PC) (Precision T7910, Dell Inc., Round Rock, TX, USA), a linear US transducer array (L7-4, Philips, Bothell, WA, USA), an HIFU transducer (5-MHz, 19-mm circular aperture, 15-mm focal distance, Precision Acoustics Ltd., Dorchester, Dorset, UK), and a programmable laser system (Phocus Mobile, Opotek Inc., Carlsbad, CA, USA) with a customized dual-output laser fiber were used in the experiment (Figure 1). The laser system worked in internal flashlamp and external Q-switch mode to ensure stable synchronization between laser firing and PA data acquisition. The flashlamp trigger output provided the primary clock signal that drove the system. In each iteration, the imaging system passively waited until the rising edge of the flashlamp trigger output was detected and then waited another 235 μs (the flashlamp to Q-switch delay value unique to the laser system for the desired power level). After that, the imaging system sent a trigger signal to the Q-switch trigger input port of the laser so that it fired and started acquiring PA data simultaneously. The laser system was programmed to scan the wavelengths from 690 nm to 950 nm with a step size of 5 nm. Because the imaging system had 64 parallel receive channels with a 2:1 multiplexing capability to receive from the 128 elements of the linear array, the laser must fire twice at each wavelength to acquire the data needed to reconstruct a complete frame (Figure 1). In order to increase the signal-to-noise ratio (SNR), 100 frames of PA data were acquired and averaged at each wavelength. A summary of the experimental conditions can be found in Table 1.

#### 2.1.2. Experimental Procedure

Ex vivo bovine tissue (sirloin) was cut into blocks (approximately 4.0 cm × 3.0 cm × 2.5 cm) as imaging specimens. The tissue block was then placed in a water bath on top of an agar backdrop at room temperature for imaging. Placing the sample on this agar block eliminates possible image artifacts due to possible reflections from the back side or light absorption in the background. The system performed the first round of MWPA and US data acquisition (Figure 2). Next, HIFU was applied with the HIFU transducer to create a lesion on the block approximately 5 mm left of the block’s centerline, and the system performed the second round of data acquisition; after that, one more lesion was created on the block at approximately its center position, and the system performed the third round of data acquisition; finally, a third lesion was created on the block approximately 5 mm right of the block’s centerline, and the system performed the fourth round of data acquisition. A 3D-printed holder held the linear array, dual-output laser fiber, and the HIFU transducer to keep their relative geometric positions fixed and to ensure that the created lesions were always on the imaging plane. The holder was attached to a linear stage (PRO165, Aerotech Inc., Pittsburgh, PA, USA) to allow for accurate component movement (Figure 2). With this data acquisition procedure, 4 frames of 53-wavelength MWPA and US data were acquired from each block of tissue, and they contained 0, 1, 2, and 3 HIFU lesions, respectively. This procedure was repeated on 19 samples, resulting in 4 × 19 = 76 MWPA images.

#### 2.1.3. Multi-Wavelength Photoacoustic Image Preprocessing

MWPA and US images were reconstructed from the acquired data with the standard delay-and-sum algorithm [52]. The image-related parameters can be found in Table 2. Each MWPA image is a 512 × 339 × 53 array, and the value of each pixel is a 53-element vector. Each element in the vector represents the PA signal intensity at the corresponding location and wavelength. For example, the 5th element in the pixel at location (1, 1) represents the intensity of PA signal at the top left corner of the image acquired at wavelength 690 + (5 − 1) × 5 = 710 nm. Likewise, each US image is a 512 × 339 array and the value of each US image pixel is a single value representing the reflected US signal intensity at the corresponding location.

The ablated tissue was cut open after the data acquisition process was finished, and then color photographs of the cross-sections of the ablated tissue were taken. By visually matching the features in the color photographs and the US images and leveraging the co-registration of the US and MWPA images, ablated and non-ablated pixels were manually labeled to generate training and test examples for the learning algorithms (Figure 3). The red pixels were labeled as ablated and the green pixels were labeled as non-ablated. The other pixels were not labeled because their status was uncertain. This uncertainty was caused by uncontrollable randomness in the ablation depth, inevitable deformation when cutting the tissue blocks, and difficulty in perfectly overlapping the cut plane with the imaging plane as well as co-registering US images with the color photographs by visual inspection. The uncertainty along the lateral direction was less significant because the lateral position of the HIFU transducer with its tight lateral focus (0.25 mm) was accurately controlled by the linear stage, which was also why the unlabeled gap area had smaller thickness in the lateral direction than the axial direction.

The amplitude of PA pressure waves $p_{0}\left(r,T,\lambda \right)$ can be calculated with the following formula [30,38]:(1)$$p_{0}\left(r,T,\lambda \right)=\Gamma \left(T\right)\mu_{abs}\left(r,\lambda \right)\Phi \left(r,\lambda \right)$$
where Γ(T) is the Grüneisen coefficient at temperature *T*, and $\mu_{abs}\left(r,\lambda \right)$ and Φ(r,λ) are the optical absorption coefficient of the tissue and the optical fluence at location r and wavelength λ, respectively.

Assuming the imaging system is linear [21], the pixel value v(r,T,λ) in the MWPA image can be written as a function of the PA pressure:(2)$$v\left(r,T,\lambda \right)=E\left(r\right)p_{0}\left(r,T,\lambda \right)+N$$
where E(r) is a constant that depends on the PA imaging system and is a function of the location r. *N* is the system noise.

The optical fluence Φ(r,λ) can be calculated as [35]:(3)Φ(r,λ)=F(λ)γ(r,λ)
where F(λ) is the optical fluence of the laser at wavelength λ measured at the output of the dual-output laser fiber and γ(r,λ) is the percentage of laser fluence at wavelength λ that can reach location r.

Combing Equations (Equation 1)–(Equation 3):(4)$$v\left(r,T,\lambda \right)=E\left(r\right)\Gamma \left(T\right)\mu_{abs}\left(r,\lambda \right)F\left(\lambda \right)\gamma \left(r,\lambda \right)+N$$

Assuming that the system noise *N* is negligible and by normalizing the pixel value v(r,T,λ) to the laser fluence F(λ):(5)$$v\left(r,T,\lambda \right)=E\left(r\right)\Gamma \left(T\right)\mu_{abs}\left(r,\lambda \right)\gamma \left(r,\lambda \right)$$

PA imaging was performed at 53 wavelengths ($\lambda_{i},i=1,2,...,53$) in the experiment (Table 1) and each MWPA pixel is a 53-element vector. Assuming that the maximum element of the pixel at r is at wavelength $\lambda_{max}$ and by normalizing each element to this maximum element:(6)$$v_{norm}\left(r,T,\lambda_{i}\right)=\frac{\mu_{abs}\left(r,\lambda_{i}\right)\gamma \left(r,\lambda_{i}\right)}{\mu_{abs}\left(r,\lambda_{max}\right)\gamma \left(r,\lambda_{max}\right)}$$

Equation (Equation 6) shows that the normalization converts the MWPA image pixel into the optical absorption spectrum which is first weighted with the optical transmission efficiency γ(r,λ) and then normalized to the peak value, eliminating the positional dependence introduced by the imaging system, the wavelength dependence introduced by the laser fluence, and the temperature dependence introduced by the Grüneisen coefficient. If one assumes that the optical transmission efficiency at a shallow depth and relatively narrow range of wavelengths has a weak dependence on the wavelength, Equation (Equation 6) reduces to a normalized absorption spectrum for a given pixel.

Based on Equations (Equation 5) and (Equation 6), we came up with our preprocessing steps for each MWPA images. First, we normalized each pixel to the laser fluence, which was measured with an energy meter (201235B, Gentec-EO, Quebec City, QC, Canada). Second, we normalized each pixel by its maximum element. The preprocessed MWPA images were fed into the segmentation algorithms introduced in the next section.

### 2.2. Lesion Segmentation

The MWPA images acquired from the 19 blocks of bovine tissue were used as training and test examples for the learning algorithms. In total, 15 were used as training examples and 4 as test examples.

#### 2.2.1. Lesion Segmentation with Traditional Machine Learning Algorithms

A total of four machine learning algorithms including quadratic discriminant analysis (QDA), neural network (NN), support vector machine (SVM), and random forest (RF) were used for lesion segmentation. Scikit-learn [53] was used to implement these algorithms.

Each pixel in the MWPA image was processed independently. Since the ablated portion was always smaller than the non-ablated portion in each MWPA image, there was an imbalance between the numbers of ablated and non-ablated pixels. Machine learning algorithms tend to produce unsatisfactory classifiers when faced with imbalanced datasets. Therefore, the non-ablated pixels were under-sampled to make the training examples balanced. For each MWPA image, all the ablated pixels were used as positive training and test examples, and an equal number of non-ablated pixels were selected randomly as negative examples. This approach eliminated the imbalance between positive and negative examples in the training data set.

Neural network, support vector machine, and random forest involved some parameters that needed to be customized. To find the optimal parameter value, the training set (15 blocks) was divided into two groups, one group (group A, 12 blocks) for training and the other group (group B, 3 blocks) for validation. Tentative models with different parameters were trained on group A and tested on group B to calculate the model’s *F*1 score [54], and the optimal model parameters were selected. *F*1 score, a commonly used metric to evaluate the performance of a classifier, is the harmonic mean of recall and precision. Assuming that in the segmentation result on a MWPA image, TP (true positive) pixels are labeled as ablated and classified as ablated, TN (true negative) pixels are labeled as non-ablated and classified as non-ablated, FP (false positive) pixels are labeled as non-ablated and classified as ablated, and FN pixels are labeled as ablated and classified as non-ablated, recall and precision are defined as:(7)$$recall=\frac{TP}{TP+FN}$$
(8)$$precision=\frac{TP}{TP+FP}$$

The *F*1 score is calculated as:(9)$$F1=\frac{2\cdot recall\cdot precision}{recall+precision}$$

The *F*1 score of a segmentation algorithm is the average value of the algorithm’s *F*1 scores on the MWPA images used as test examples.

The neural network contained three layers: an input layer, a hidden layer, and an output layer. Neural networks with different numbers of hidden layer nodes were trained on group A and tested on group B to check how the *F*1 score would change as the number of hidden layer nodes increased. Based on the *F*1 score vs. number of hidden layer nodes curve (Figure 4a), the number of hidden layer nodes was set to 53 because increasing this number further did not increase the *F*1 score significantly and would consume more resources and increase the run time.

The support vector machine used the default radial basis function kernel. Models with different values for the penalty term C [55] were trained on group A and tested on group B to check how the *F*1 score would change as term C increased. Based on the *F*1 score vs. log2(C) curve (Figure 4b), C was set to 16 because the *F*1 score was close to its maximum value and increasing C further resulted in significantly longer run time.

Random forest classifiers with different numbers of trees were trained on group A and tested on group B to check how the *F*1 score would change as the number of trees increased. Based on the *F*1 score vs. number of trees curve (Figure 4c), the number of trees was set to 50 because the *F*1 score was close to its maximum value and increasing the number of trees further would consume more resources and increase the run time. Then, the number of trees was fixed, and random forest classifiers with different “minimum number of samples in each leaf node” (min_samples_leaf [53]) were trained on group A and tested on group B. The *F*1 score decreased (by approximately 5%) as the min_samples_leaf value increased from 100 to 5000 (Figure 4d), and so does the maximum tree depth (by approximately 20) (Figure 4e). In order to prevent overfitting while retaining good performance, the min_samples_leaf value was set to 1000.

#### 2.2.2. Lesion Segmentation with Convolutional Neural Network

The CNN was composed of 6 layers (Figure 5). Layers 1-3 were composed of a convolutional layer followed by rectified linear unit (ReLU) activation [56] and a max-pooling layer. In the convolutional layer, the 53-channel input image was convolved with a 53-channel 3 × 3 convolution kernel (Table 3) to generate a single-channel output of the same size as the input (Figure 6). This operation was repeated with 53 different kernels to generate a 53-channel output, and then ReLU activation function was applied to each pixel in the output. In the max-pooling layer, a 2×2 window slided over each channel with a step size of 2 in each dimension (Table 3), and the maximum value in the window was selected as the value in the output of the max-pooling layer. With such an operation, each max-pooling layer halved the image size without changing the number of channels.

Layers 4 and 5 did not contain max-pooling layers and their convolution kernel size was 1 × 1 (Table 3), so layers 4 and 5 were equivalent to a pixel-wise fully connected neural network. 53 convolution kernels were used in the convolutional layer in layer 4 and 2 were used in layer 5 to generate a 2-channel output. Layer 6 was a bilinear upsampling layer that interpolated the 2-channel output from layer 5 to the same size as the input MWPA image. Each channel in the output of layer 6 was a map of scores that reflected how likely each pixel was to belong to the class that the channel corresponded to. The argmax function was applied to the output of layer 6 to generate the final segmentation result. The CNN was implemented with TensorFlow [57].

The CNN was trained by minimizing a loss function. The loss function calculated a weighted average cross entropy between the output of the network and the human labeling. For non-ablated pixels, the weight was 1. For ablated pixels, the weight was the ratio of the number of non-ablated to ablated pixels. For unlabeled pixels, the weight was 0. This strategy helped minimize the impact of the imbalance between ablated and non-ablated pixels. An Adam optimizer [58] was used for training the CNN for 200 epochs.

Data augmentation was used to generate more training examples for the CNN because it required significantly more training examples than traditional machine learning algorithms. Each image was blurred with a 3 × 3, 5 × 5, and 7 × 7 window and also corrupted with 2 versions of random Gaussian noise (Figure 7). Additionally, the image was flipped upside down, left to right, and rotated by 180°. In this way, the number of examples was multiplied by 24, resulting in 15 blocks × 4 frames/block × 24 = 1440 MWPA images for training.

#### 2.2.3. Wavelength Selection

Scanning 53 wavelengths with a 20-Hz repetition-rate laser system would result in a frame rate smaller than 0.5 frames per second (FPS), even when system channels could address all transducer elements concurrently and no signal averaging is employed. To facilitate real-time implementation, wavelength selection was conducted to reduce the number of wavelengths that were needed to segment HIFU lesions in MWPA images. The built-in sequential feature selection function in MATLAB (The MathWorks, Inc., Natick, MA, USA) [59] was used for wavelength selection. The ablated and non-ablated pixel data were shuffled and divided into two groups: one for training (90%) and the other for testing (10%). The algorithm started with an empty wavelength set. In each iteration, the algorithm added each one of the unselected wavelengths to the wavelength set, fit the training data to a quadratic classifier, and calculated its error rate on the test data. The algorithm then selected the wavelength that produced the minimum error rate, added it to the wavelength set, and stepped into the next iteration. This process continued until the error rate stopped decreasing. The quadratic classifier was chosen for this step because it was simple and trained quickly. Since the wavelength selection algorithm took a greedy approach, the earlier-selected wavelengths were more “important”. The learning algorithms were trained and tested with the first 5, 4, 3, and 2 most important wavelengths to evaluate the performance as the number of wavelengths decreased. The inference time (time needed for an algorithm to produce the segmentation result for one MWPA image) of each algorithm for a single frame was also measured for each wavelength count on a PC (Precision T5610, Dell Inc., Round Rock, TX, USA) with two 16-core central processing units (CPU) (Intel Xeon processor E5-2650 v2 @ 2.6 GHz, Intel Corporation, Santa Clara, CA, USA).

## 3. Results

### 3.1. Lesion Segmentation with All Wavelengths

The test results show that all the learning algorithms are able to capture the major part of the lesion (Figure 8). The results of the traditional machine learning algorithms appear significantly noisier than the CNN. The *F*1 score of each algorithm is calculated for the test examples (row 1, Table 4). The *F*1 scores of traditional machine learning algorithms are mostly around 90%. The *F*1 score of the CNN is close to 100%, which means that it achieves almost 100% precision and recall. The test results of the CNN on the four MWPA images from the same block of bovine tissue clearly show the process of lesions being laterally extended after each application of HIFU (Figure 9 and Figure 10). The lesion boundaries were generated with the same method as in [35].

### 3.2. Lesion Segmentation with Reduced Wavelengths

The sequential feature selection algorithm selected 46 wavelengths, which include every wavelength except 695, 700, 705, 755, 765, 800 and 915 nm. As the number of wavelengths increased from 1 to 46, the error rate went down from 27.36% to 14.69% (Figure 11). To examine how the wavelengths selected by the automated feature selection algorithm relate to the PA spectra of the ablated and non-ablated pixels, we calculated an average normalized spectrum for both types of pixels. To do this, we first normalized the measured PA spectrum by the laser energy spectrum for each class of pixels that were manually labeled. Then the spectrum for each pixel was normalized to its own maximum resulting in a normalized representation of optical absorption for each pixel. The spectra for ablated and non-ablated pixels were then averaged (Figure 12a). The ratio of the spectra was used to depict in what range the spectra of the ablated and non-ablated pixels differ most significantly (Figure 12b). The five most “important” wavelengths in lesion segmentation reported by the feature selection algorithm are 780, 775, 785, 790, and 720 nm (sorted from most to least important). At these wavelengths, we observe that the ratio of the average normalized ablated pixel value to the average normalized non-ablated pixel value is at or close to the maximum. We chose the PA images at these wavelengths for our reduced-wavelength MWPA images for lesion segmentation. These MWPA images were preprocessed the same way as the 53-wavelength MWPA images before being processed by the segmentation algorithms.

With 5 wavelengths instead of 53, the segmentation results of all the traditional machine learning algorithms are noisier (Figure 13). The noise level of the results from the CNN is not apparently higher, but the shape of its lesion segmentation appears less natural. The *F*1 scores of all the learning algorithms decreased (row 2, Table 4). The *F*1 scores of the traditional machine learning algorithms decreased by between 4.80% and 7.20% while the *F*1 scores of the CNN decreased by less than 3.40% (row 1, Table 5). With 5 wavelengths, the test results of the CNN on the four MWPA images from the same block of bovine tissue still clearly show the process of lesions being laterally extended after each application of HIFU (Figure 14 and Figure 15).

As the number of wavelengths continued decreasing, the *F*1 scores continued decreasing in general (row 3–5, Table 4). For the traditional machine learning algorithms, one fewer wavelength caused between 0.13% and 4.61% drop (row 2–4, Table 5). As the number of wavelengths became smaller, the *F*1 score decreased at a higher rate. For the CNN, one fewer wavelength caused between 0.01% and 3.81% drop (row 2–4, Table 5). The *F*1 score of the CNN with 2 wavelengths is higher than the traditional machine learning algorithms with 53 wavelengths, and the lesions can be clearly segmented with lower noise level (Figure 16).

## 4. Discussion

### 4.1. High-Performance Lesion Segmentation

The CNN performs well in segmenting HIFU lesions in MWPA images. With 53 wavelengths, the *F*1 score is almost 100%. Even when the number of wavelengths is reduced to 2, the *F*1 score is still higher than 92%. The performance of the traditional machine learning algorithms is good with 53 wavelengths, but decreases significantly and more quickly than the CNN as the number of wavelengths decreases. This is because the traditional machine learning algorithms process each pixel independently and thus the segmentation results are purely based on the optical absorption spectrum information. Meanwhile in the CNN, the convolutional layers capture not only the optical absorption spectrum but also the 2D textural features in the MWPA images. This also explains why some side lobes of the lesions are classified as ablated in traditional machine learning algorithms yet classified as non-ablated in the CNN. In MWPA image preprocessing, most of these side lobe pixels are not labeled because they are close to the lesion boundaries and have uncertain status. Traditional machine learning algorithms tend to classify these pixels as ablated because their optical absorption spectrum is consistent with the lesions from which the side lobes are derived, while the CNN is less likely to classify these pixels as ablated because their 2D textural features could differ from real ablated pixels.

The CNN architecture used in this study was inspired by the fully convolutional network in [40], which is the basis for the current state-of-the-art deep learning semantic segmentation techniques [60]. A more sophisticated neural network could be used to further improve the lesion segmentation performance if more training examples were available.

### 4.2. Convenient Implementation

With off-the-shelf machine/deep learning libraries, lesion segmentation in MWPA images is a convenient software development task. The learning algorithms are trained end-to-end, and no extra processing is required to extract “deeper” features from the MWPA images. In our lesion segmentation task, most of the development effort was spent tuning the parameters of the learning algorithms to find the optimal values. However, some learning algorithms (e.g., CNN, support vector machine) require longer training times and thus it is difficult to identify parameter values that are comprehensively optimal. Further data analysis is necessary in order to extend beyond the application level and explore what features distinguish ablated and non-ablated tissue apart.

### 4.3. Advantages over Temperature-Measurement-Based Methods

HIFU lesion formation can also be monitored by measuring temperature in the image view. MRI has superb temperature-mapping capability and is widely used for temperature measurement during HIFU therapies. Due to the linear dependence of the PA signal amplitude on temperature and the linearity of the PA imaging system, temperature can be measured indirectly from PA imaging with proper calibration [21]. A drawback of this temperature-measurement-based approach is that the temperature change is transient. This means that it can segment HIFU lesions only when the HIFU therapy is on, and it is not possible to repeat the segmentation after the therapy is completed. The approach we proposed in this work is based on the optical absorption spectrum and the 2D textural information, which change permanently during HIFU therapies. Therefore, the segmentation can be performed during the HIFU therapy as well as after its completion. Another drawback of the temperature-measurement-based approach with PA imaging is that it needs calibration, which requires temperature data. In experimental settings, the data can be acquired by inserting extra devices (e.g., thermocouples) into the tissue to measure the temperature. However, this is challenging to perform in vivo. Furthermore, this insertion introduces unpredictable exogenous factors that could affect the linear relation between the temperature and the PA signal amplitude in the tissue, resulting in suboptimal calibration results. The approach we proposed does not require additional devices to be inserted into the tissue for calibration. So, this approach is more reliable and easier to be validated.

### 4.4. Potential for Real-Time Implementation

When running on a PC with two 16-core CPUs, the inference time of quadratic discriminant analysis, neural network, random forest, and CNN is less than one second (Table 6). So, all of the presented methods can process more than one frame within a second. As the number of wavelengths decreases, the inference time of quadratic discriminant analysis, neural network, random forest, and CNN also decreases. When the number of wavelengths is 2, the inference time of the CNN is 157.38 ms, which corresponds to a frame rate of 6.35 FPS, while the maximum frame rate for the current imaging system used in this study is 5 FPS (20 repetition/second ÷ 2 repetition/wavelength ÷ 2 wavelength/frame) when no signal averaging was employed. So, the computational power of the PC is potentially sufficient to implement an MWPA imaging system with the CNN running in real time. With accelerating hardware such as a graphics processing unit (GPU) [61] or a field-programmable gate array [62], the inference time will be further decreased.

Compared with the other algorithms, the inference time of quadratic discriminant analysis is significantly shorter because it assumes that the distribution of each category of pixels is a single Gaussian distribution, and this model is computationally simpler. The inference time of support vector machine is significantly longer. This is because its inference time complexity is O(n^3^) [63] where n is the number of training examples. The inference times of quadratic discriminant analysis, neural network, random forest, and CNN do not change with n.

In our experiment, a commercial transducer probe (L7-4) is used for PA data acquisition. In order to increase the SNR, multiple (100) acquisitions are averaged for each frame. A 2D CMUT with integrated electronics has been shown to achieve high SNR in PA imaging [64] and is a good alternative to commercial piezoelectric probes for HIFU lesion segmentation in real-time MWPA imaging. For the presented application in this study, we developed a similar system capable of HIFU ablation, and real-time ultrasound and photoacoustic imaging [65].

### 4.5. Potential Improvement with Gold-Standard Training Examples

The MWPA images were not fully labeled since we labeled the ablated pixels manually mainly by visual inspection of the tissue cross-section after completing the experiment. With this information, it is not possible to label every pixel with complete confidence. The difficulty lies not only in distinguishing ablated from non-ablated tissue in the tissue picture but also in co-registering the tissue picture with the PA and US images since deformation is inevitable when cutting the tissue blocks. Furthermore, we could not guarantee that the cut plane perfectly overlaps with the imaging plane. An imaging modality that can provide a gold-standard reference (such as MRI) on lesion segmentation would improve the experiment and provide stronger validation of our approach.

## 5. Conclusions

In this paper, we presented a CNN-based approach to segment HIFU lesions in MWPA images. Traditional machine learning algorithms were also trained and tested to compare with the CNN-based approach, and the results show that the performance of CNN significantly exceeds traditional machine learning algorithms. Feature selection was conducted to reduce the number of wavelengths to facilitate real-time implementation. The performance of the CNN with 2 wavelengths exceeds the traditional machine learning algorithms with 53 wavelengths. This work demonstrates the high performance of CNN in HIFU lesion segmentation and the potential to operate it in real time. Further work includes running the CNN with accelerating hardware to assess its capability in monitoring HIFU lesion formation in real time and potentially in vivo, as well as further improving the segmentation performance with more sophisticated neural networks.

## Figures and Tables

**Figure 1 bioengineering-10-01060-f001:**
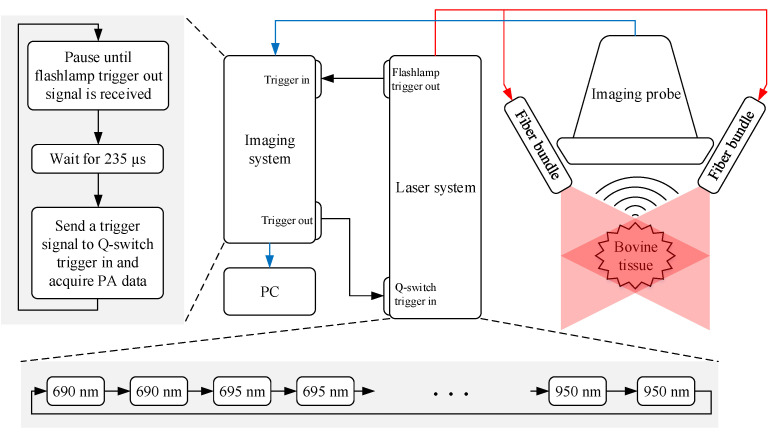
Experimental setup for multi-wavelength photoacoustic characterization of HIFU-induced lesions.

**Figure 2 bioengineering-10-01060-f002:**
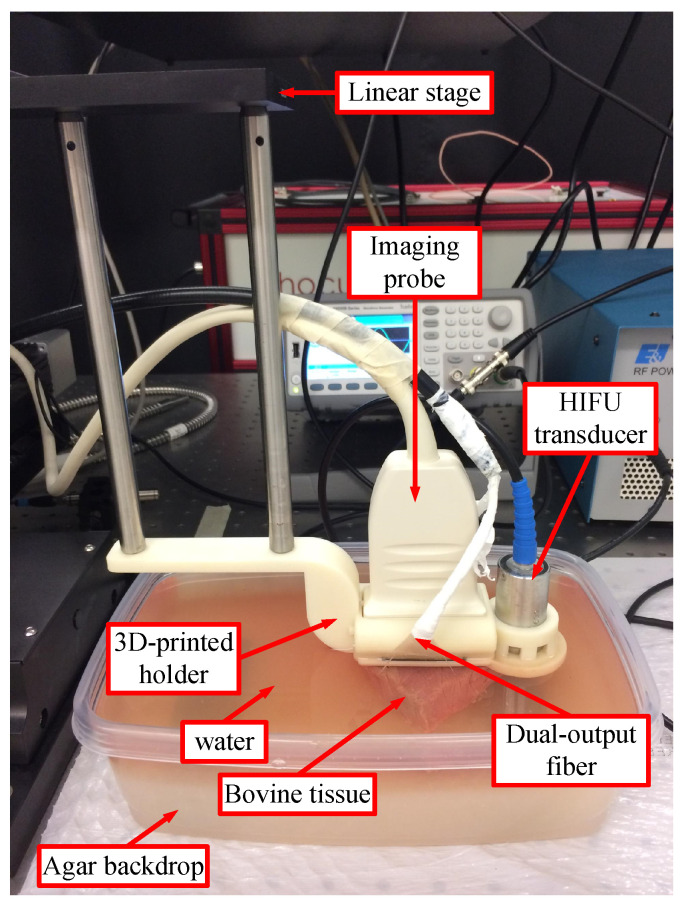
Imaging components held by the 3D-printed holder attached to the linear stage.

**Figure 3 bioengineering-10-01060-f003:**
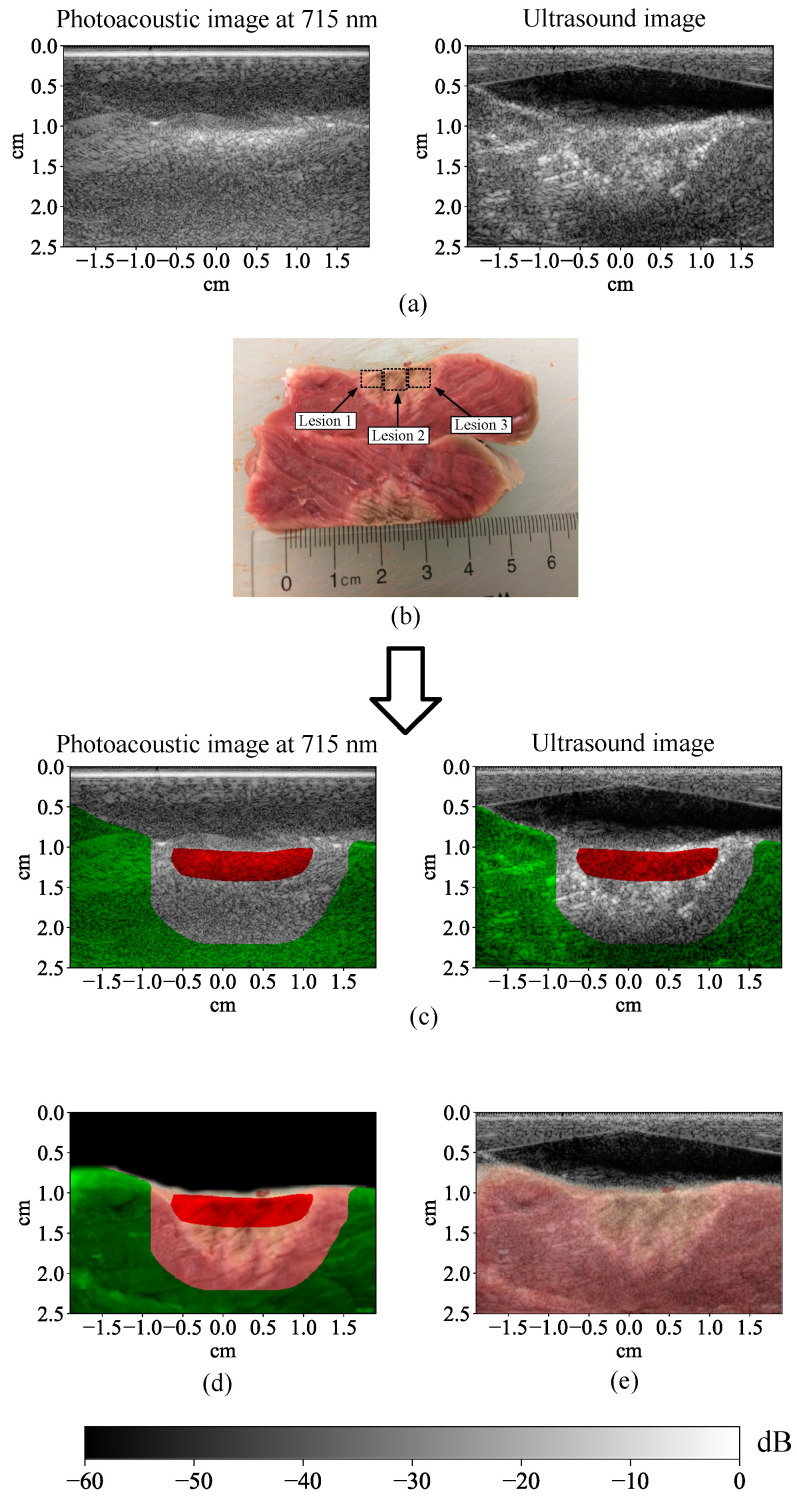
(**a**) Photoacoustic and ultrasound images of the bovine tissue with three lesions created. (**b**) Color photograph of the cross-section of the bovine tissue after ablation. (**c**) Manual labeling on top of the photoacoustic and ultrasound images. (**d**) Manual labeling on top of the color photograph of the cross-section of the bovine tissue. (**e**) Co-registration of the photograph of the bovine tissue with the ultrasound image (video: https://drive.google.com/file/d/16MRtYhNZHpTM5PQdD1bMAnRmhPvVT0Do/view?usp=share_link, accessed on 30 December 2022). The colorbar is valid for (**a**,**c**,**e**).

**Figure 4 bioengineering-10-01060-f004:**
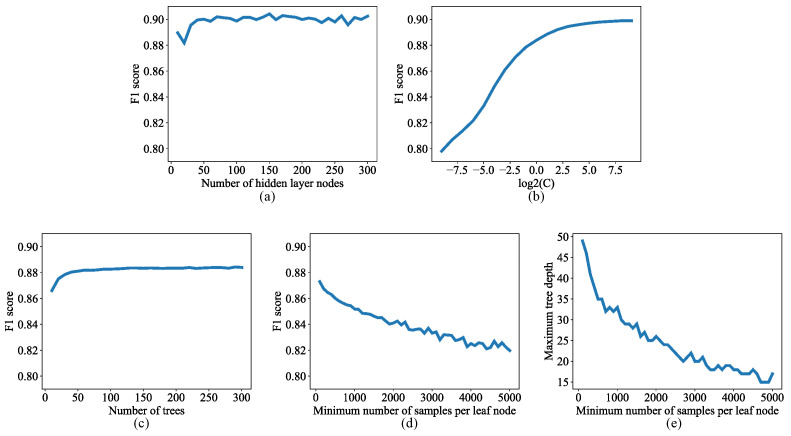
(**a**) *F*1 score vs. number of nodes in the hidden layer of the neural network. (**b**) *F*1 score vs. log_2_(C) in the support vector machine. (**c**) *F*1 score vs. number of trees in the random forest. (**d**) *F*1 score vs. minimum number of samples per leaf node in the random forest. (**e**) Maximum tree depth vs. minimum number of samples per leaf node in the random forest.

**Figure 5 bioengineering-10-01060-f005:**
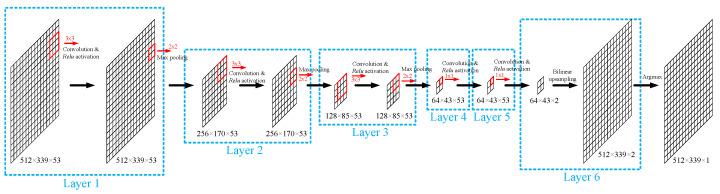
Architecture of the convolutional neural network. The size of each image is described as width × height × depth (number of channels). The size of each kernel is described as width × height.

**Figure 6 bioengineering-10-01060-f006:**
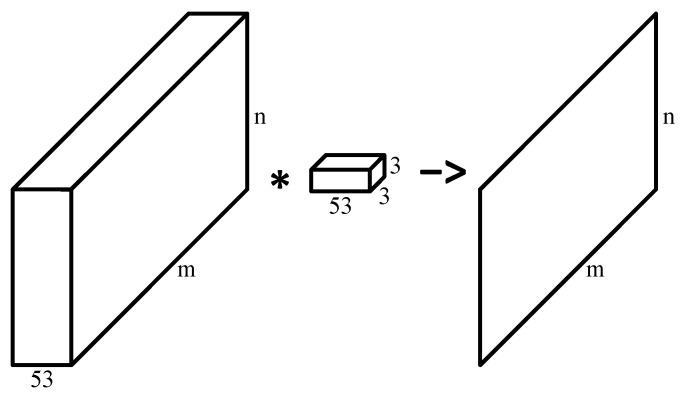
Convolutional layer in layer 1–3.

**Figure 7 bioengineering-10-01060-f007:**
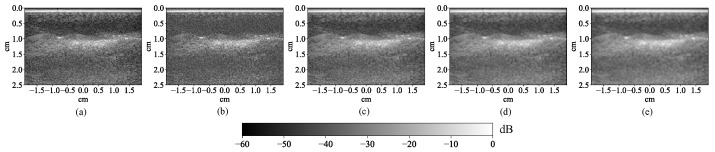
An example for data augmentation on a photoacoustic image: (**a**) Original image. (**b**) Image corrupted with Gaussian noise. (**c**) Image blurred with a 3 × 3 window. (**d**) Image blurred with a 5 × 5 window. (**e**) Image blurred with a 7 × 7 window.

**Figure 8 bioengineering-10-01060-f008:**
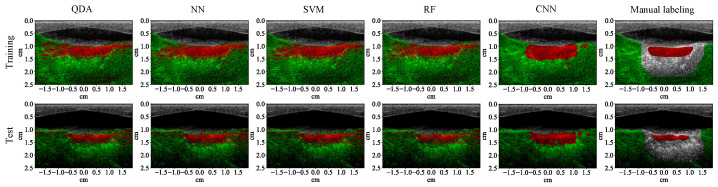
Segmentation results (with 53 wavelengths) superimposed on the ultrasound image for a training (row one) and test (row two) example that contains 3 lesions. Each column except the last one corresponds to a specific learning algorithm (from left to right: quadratic discriminant analysis (QDA), neural network (NN), support vector machine (SVM), random forest (RF), and convolutional neural network (CNN)), the last column shows the manual labeling for training and measuring the segmentation performance.

**Figure 9 bioengineering-10-01060-f009:**
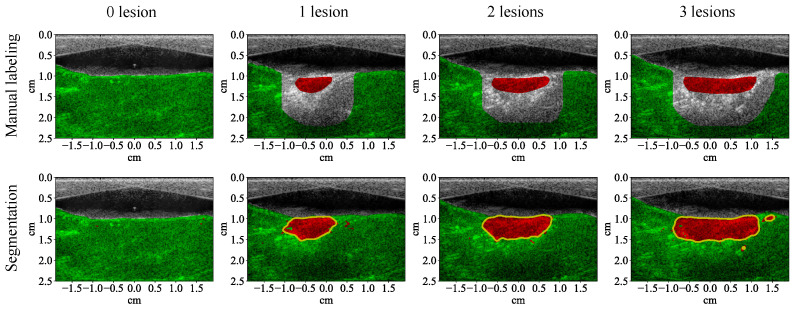
Segmentation results (with 53 wavelengths) of the convolutional neural network superimposed on the ultrasound image showing lesion formation for a training example (row two). Each column corresponds to a different number of lesions formed (from left to right: 0, 1, 2, 3). Row one shows the manual labeling for training.

**Figure 10 bioengineering-10-01060-f010:**
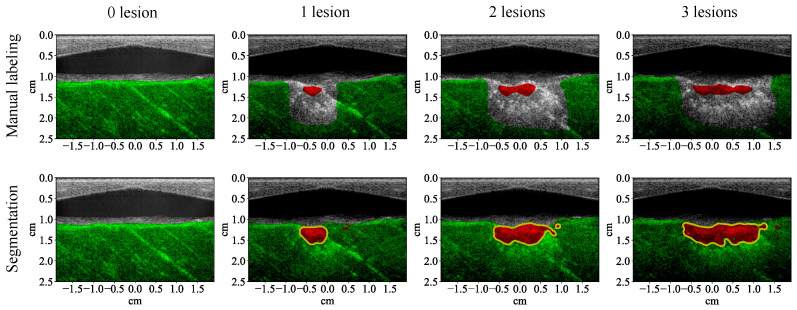
Segmentation results (with 53 wavelengths) of the convolutional neural network superimposed on the ultrasound image showing lesion formation for a test example (row two). Each column corresponds to a different number of lesions formed (from left to right: 0, 1, 2, 3). Row one shows the manual labeling for measuring the segmentation performance.

**Figure 11 bioengineering-10-01060-f011:**
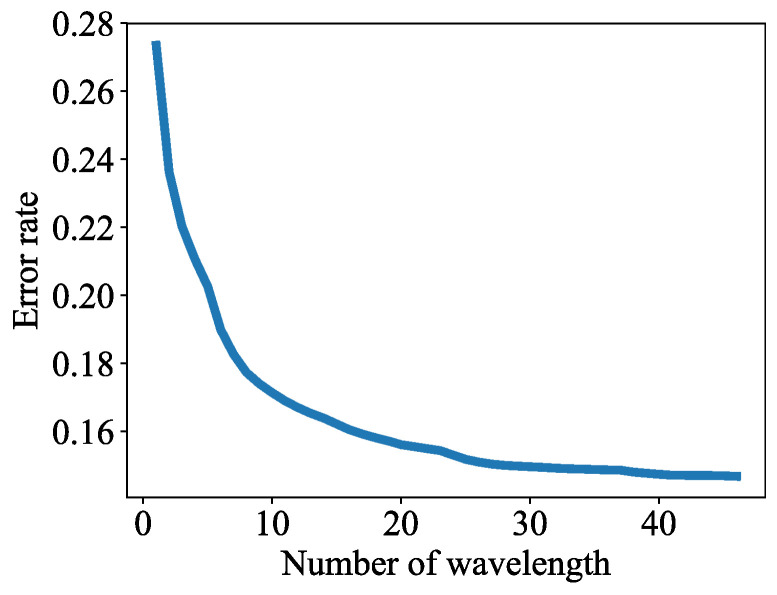
Error rate vs. number of wavelengths in feature selection.

**Figure 12 bioengineering-10-01060-f012:**
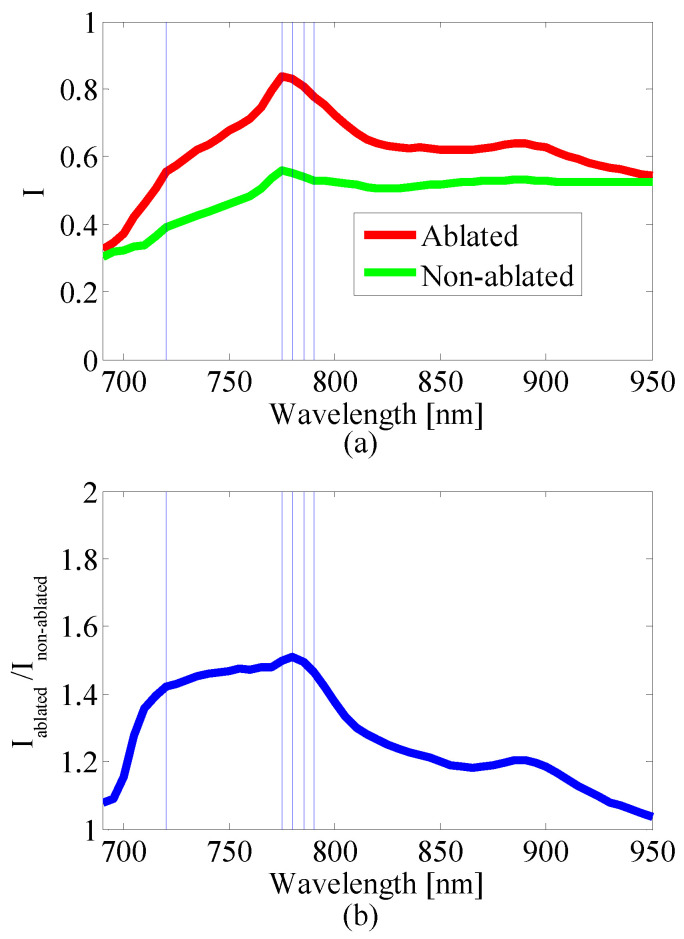
(**a**) The average normalized ablated and non-ablated pixel value as a function of wavelength. (**b**) The ratio of average normalized ablated to average normalized non-ablated pixel value as a function of wavelength. Positions of the five vertical lines (five most “important” wavelengths) from left to right: 720, 775, 780, 785, and 790 nm.

**Figure 13 bioengineering-10-01060-f013:**
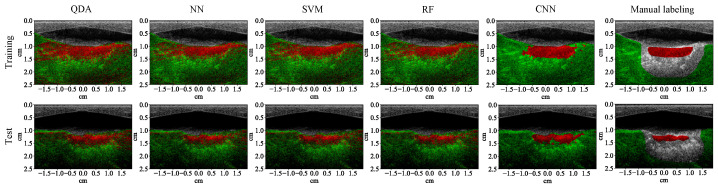
Classification results (with 5 wavelengths) superimposed on the ultrasound image for a training (row one) and test (row two) example that contains 3 lesions. Each column except the last one corresponds to a specific learning algorithm (from left to right: quadratic discriminant analysis (QDA), neural network (NN), support vector machine (SVM), random forest (RF), and convolutional neural network (CNN)), the last column shows the manual labeling for training and measuring the segmentation performance.

**Figure 14 bioengineering-10-01060-f014:**
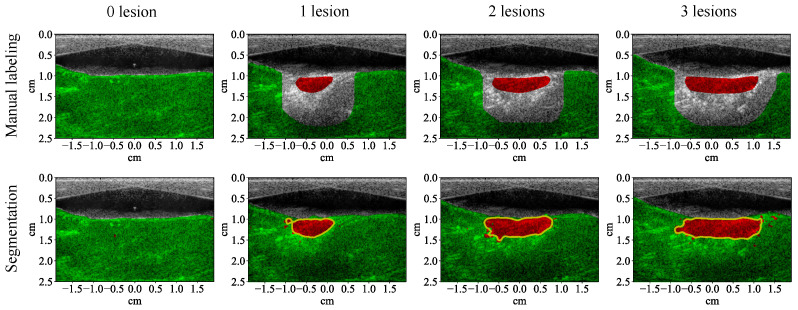
Segmentation results (with 5 wavelengths) of the convolutional neural network superimposed on the ultrasound image showing lesion formation for a training example (row two). Each column corresponds to a different number of lesions formed (from left to right: 0, 1, 2, 3). Row one shows the manual labeling for training.

**Figure 15 bioengineering-10-01060-f015:**
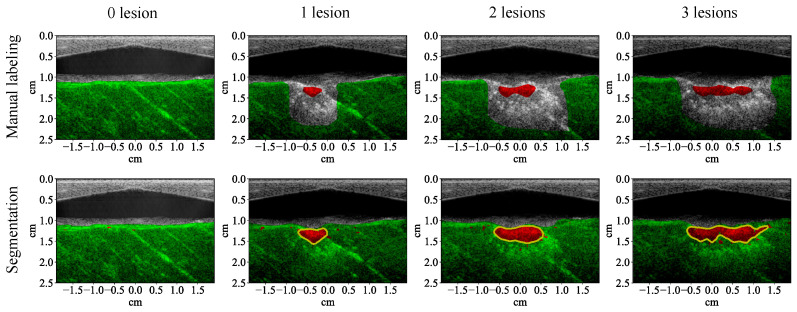
Segmentation results (with 5 wavelengths) of the convolutional neural network superimposed on the ultrasound image showing lesion formation for a test example (row two). Each column corresponds to a different number of lesions formed (from left to right: 0, 1, 2, 3). Row one shows the manual labeling for measuring the segmentation performance.

**Figure 16 bioengineering-10-01060-f016:**
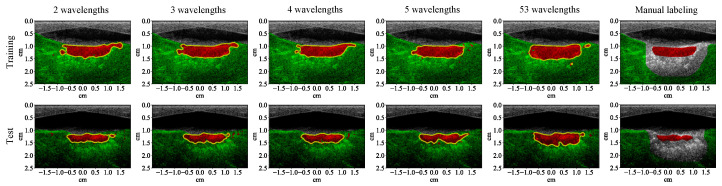
Segmentation results of the convolutional neural network superimposed on the ultrasound image for a training (row one) and test (row two) example that contains 3 lesions. Each column except the last one corresponds to a different number of wavelengths used in segmentation (from left to right: 2, 3, 4, 5, 53), the last column shows the manual labeling for training and measuring the segmentation performance.

**Table 1 bioengineering-10-01060-t001:** Experimental Conditions.

Linear array:	
Center frequency	5.2 MHz
Number of elements	128
Size (azimuth)	38 mm
Imaging system:	
Sampling rate	62.5 MS/s
Number of receive channels	64
Multiplexing capability	2:1
Number of acquisitions for average	100
Laser system:	
Pulse width	5 ns
Repetition rate	20 Hz *
Imaging wavelength	690 nm to 950 nm
Scanning step size	5 nm
Number of wavelengths	53
Maximum pulse energy	14.74 mJ (715 nm) **
Minimum pulse energy	7.93 mJ (950 nm) **
Dual-output fiber:	
Aperture (per branch)	0.25 mm × 40 mm

* This value refers to the repetition rate of the laser. The ideal single-wavelength photoacoustic data acquisition rate of the system is 20 firing/second ÷ 2 firing/frame = 10 frames per second. Its actual value turned out to be approximately 7 frames per second due to the overhead associated with the file I/O operations. ** The energy values were measured at the output of the dual-output fiber.

**Table 2 bioengineering-10-01060-t002:** Imaging Parameters.

Imaging frequency	5.2 MHz
Distance between pixels	73.92 μm ($\frac{1}{4}$ wavelength)
Size of the image view	3.78 (width) cm × 2.5 (height) cm
Number of pixels	512 (width) × 339 (height)

**Table 3 bioengineering-10-01060-t003:** Convolutional Neural Network Parameters.

Layer 1–3:	
Convolution kernel size	3 × 3
Convolution stride	1 × 1
Activation function	*relu*
Max-pooling kernel size	2 × 2
Max-pooling stride	2 × 2
Layer 4–5:	
Convolution kernel size	1 × 1
Convolution stride	1 × 1
Activation function	*relu*

**Table 4 bioengineering-10-01060-t004:** *F*1 Score (%) for Machine Learning Algorithms.

Number of Wavelengths	QDA	NN	SVM	RF	CNN
53	91.34	92.62	92.65	89.93	99.63
5	84.77	85.42	85.51	85.13	96.26
4	84.64	84.68	84.76	84.67	94.05
3	82.71	82.77	83.08	82.60	96.60
2	78.45	78.16	78.74	78.18	92.79

**Table 5 bioengineering-10-01060-t005:** *F*1 Score (%) Decrease for Machine Learning Algorithms.

Number of Wavelengths	QDA	NN	SVM	RF	CNN
from 53 to 5	6.57	7.20	7.14	4.80	3.37
from 5 to 4	0.13	0.74	0.75	0.46	2.21
from 4 to 3	1.93	1.91	1.68	2.07	−2.55
from 3 to 2	4.26	4.61	4.34	4.42	3.81

**Table 6 bioengineering-10-01060-t006:** Inference Time (millisecond) for Machine Learning Algorithms.

Number of Wavelengths	QDA	NN	SVM	RF	CNN
53	377.99	357.57	>1000	322.55	504.20
5	45.96	127.61	>1000	307.75	170.21
4	42.29	118.73	>1000	303.97	163.19
3	34.99	112.87	>1000	271.98	161.82
2	33.65	108.06	>1000	248.31	157.38

## Data Availability

Not applicable.

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
