# Peer review of "Deep-Learning-Based High-Intensity Focused Ultrasound Lesion Segmentation in Multi-Wavelength Photoacoustic Imaging"

_bioengineering, 2023, doi:10.3390/bioengineering10091060_

Round 1

Reviewer 1 Report

In this manuscript, the authors propose a deep learning-based method to segment HIFU lesions in MWPA images, which realizes pixel-level classification in MWPA imaging, helps doctors monitor HIFU lesions, and will make an important contribution to the realization of fully automated HIFU treatment. Compared with traditional machine learning methods, CNN reduces the wavelength and still maintains good real-time segmentation performance. Compared with the commonly used MRI monitoring and imaging modality, the combination of ultrasound and photoacoustic monitoring is not only less costly, but also more compatible with HIFU devices. This manuscript provides a feasible approach for better monitoring of HIFU therapy. Therefore, I suggest the authors to add content to clarify the obscurity before publication.

1. The manuscript is a bit short on comparisons between MWPA and other imaging monitoring modalities, could there be more as well as more detailed comparisons between them to suddenly make the advantages of MWPA more obvious and specific?

2. The manuscript only uses a self-built CNN to compare with traditional machine learning algorithms, but does not compare the self-built CNN with the current commonly used semantic segmentation networks, perhaps these would have better results.

Reviewer 2 Report

The authors utilized two approaches CNN  and machine learning algorithms. It seems to use an intensity value as a feature. Please clarify that in your manuscript. For the Feature reduction method, which one you used it? Explain that in detail. Which features you discarded and which ones you kept it?

How did you train your created CNN, and What are your options (Optimizers, epochs, validation set).

For machine learning that data was imbalanced you did not clarify the process in your paper, how did you deal with imbalanced data?

Did you supply us with a graphical abstract?

Expand your conclusion to include more details. 

Mention the nature of your collected dataset—the number of images, The similarity index with ground truth. 

Reviewer 3 Report

This report addresses the use of multiple wavelength photoacoustic (PA) imaging to detect the lesions in bovine tissue by HIFU. The authors build up a set-up to take the PA images. According to the results, it is found the CNN network with only 2 wavelengthes would achieve higher accuracy than the other three machine learning methods. I would give my suggestions for more improvements on this paper.
1. It seems that the main task was the use of CNN network and conventional machine learning texhnques. It is seldom reviewed in Sec. Introduction to show reasons why the authors favored CNN with MWPA technique. And the part describing HIFU is too much in Sec. Introduction.

2. The role of HIFU in this study is advised to be described in more details. Was HIFU in this study used to create lesions or just for observing and detecting lesions. It's not very clear described in Secs. 1 and 2.

3. The description for the selected maching learning techniques are also not enough. Are there tuning parameters required in executing these ML techniwues?

4. The English in this paper is good. Checking the use of words and making this paper by plain English would be friendly to readers to read this paper smoothly.

The statements are expressed correctly. However, adaquite wording is advised with plain English is advised for readers for smooth comprehension.

Round 2

Reviewer 2 Report

The paper is clarified